# Effect of Cooling Rate on Microstructure of In Situ TiC-Reinforced Composite Surface Layers Synthesized on Ductile Cast Iron by Laser Alloying

**DOI:** 10.3390/ma17040932

**Published:** 2024-02-17

**Authors:** Damian Janicki, Artur Czupryński, Jacek Górka, Krzysztof Matus

**Affiliations:** 1Department of Welding, Silesian University of Technology, Konarskiego 18A, 44-100 Gliwice, Poland; artur.czuprynski@polsl.pl (A.C.); jacek.gorka@polsl.pl (J.G.); 2Materials Research Laboratory, Silesian University of Technology, Konarskiego 18A, 44-100 Gliwice, Poland; krzysztof.matus@polsl.pl

**Keywords:** laser surface alloying, titanium carbide, in situ composite, cementite, ductile cast iron, thermography, cooling rate

## Abstract

The microstructure of the in situ TiC-reinforced composite surface layers developed during laser surface alloying of a ductile cast iron substrate with titanium was related to the solidification conditions in the molten pool. The solidification conditions were estimated using infrared thermography. It was found that the cooling rates of the melt up to about 700 °C/s enable the complete reaction between carbon and the entire amount of titanium introduced into the molten pool. In turn, the cooling rate of about 280 °C/s for the melt containing 8.0 wt% Ti allows the TiC particles to grow in the dendritic form with well-developed secondary arms and a total size of up to 30 µm. For a constant Ti content, the cooling rate of the melt had no effect on the TiC fraction. The increase in the cooling rate elevated the retained austenite fraction in the matrix material, lowering its hardness.

## 1. Introduction

Fe-based metal matrix composites (MMCs) reinforced by ceramic particles are among the most common materials used for the fabrication of wear-resistant components [1,2,3,4,5]. There has been considerable research on the fabrication of Fe-based MMC coatings reinforced by in situ synthesised TiC particles in the last two decades [6,7,8]. Several approaches have been developed to fabricate such in situ composite structures on the working surfaces of machine parts made of different cast iron grates [9,10,11,12,13]. The laser surface alloying (LSA) process is particularly promising for the enhancement of wear performances of cast iron components [14]. The precise control of the solidification conditions in the molten pool and the ability to treat localised areas make the LSA process a unique method for the production of locally selected composite surface layers on complex-shaped machine parts [15]. Additionally, the high carbon content in the cast iron substrates ensures excellent conditions for the in situ formation of TiC reinforcing precipitates [16].

The growth morphology and volume fraction of reinforcing particles play a key role in wear behaviour of the in situ MMC composites [17]. Furthermore, the wear characteristics of MMC materials are highly influenced by the mechanical properties of the matrix material [18]. In turn, the microstructure of the in situ MMC surface layers fabricated via laser processing is affected by both the solidification conditions in the molten pool and the composition of the melt [19]. Therefore, to make it possible to tailor the wear performances of the in situ TiC-reinforced Fe-based composite surface layers for a given industrial application, an understanding regarding the role of the thermal conditions in the molten pool in the microstructure’s development is necessary.

Unfortunately, to date, few detailed investigations of the influence of the thermal conditions in the molten pool on the microstructure of TiC-reinforced Fe-based layers produced via laser surface treatment processes have been reported. Emamian et al. [20] used the numerical simulation of the laser cladding process to calculate the thermal conditions in the molten pool during the fabrication of Fe-Ti-C alloy system coatings. They compared the numerical results with experimental data and suggested that the morphology and size of the in situ formed TiC particles strongly depend on the cooling rate in the molten pool. Additionally, they reported that the change in the dilution level and the resulting change in the melt composition affect the growth morphology of TiC particles.

In previous works [14,21], the results of the thermodynamic calculations and microstructural analysis of TiC-reinforced surface layers fabricated on ferritic–pearlitic ductile cast iron via the LSA process with pure titanium powder were reported. The results indicated that, due to a very low Gibbs free energy for the formation of the TiC phase, the fraction of TiC reinforcing particles in the alloyed layers was directly dependent on the titanium concentration in the molten pool, regardless of the change in the solidification conditions. Moreover, it has been found that the fraction of the retained austenite is affected by the traverse speed, which in turn significantly determines solidification conditions in the molten pool. The current work extends the research programme presented in earlier works with an emphasis on determining the effect of the solidification conditions in the molten pool on the fraction and morphology of TiC precipitates and the phase composition of the matrix.

## 2. Materials and Methods

### 2.1. Materials

Rectangular test plates with the dimensions 60 mm length × 25 mm width × 10 mm thick were fabricated from a ductile cast iron (DCI) grate EN-GJS-700-2. The chemical composition of the used DCI is given in Table 1. The surfaces of the test plates were ground to produce a smooth finish and cleaned with acetone prior to laser processing. A commercial purity titanium powder (AMPERIT 154 H.C. Starck GmbH, Munich, Germany), with a particle size range of 40–70 μm, was selected as the alloying material. The chemical compositions of the DCI plates and the alloyed beads were measured using a LECO GDS 500A optical emission spectrometer and a LECO CS125 Carbon Analyser (LECO Corporation, St. Joseph, MI, USA).

### 2.2. Laser Processing

The surface alloying trials were performed using an experimental stand equipped with a 2.0 kW continuous-wave high-power direct-diode laser (HPDDL, Rofin DL020 Rofin-Sinar Laser GmbH, Hamburg, Germany). The HPDD laser used had a rectangular beam with a top-hat intensity distribution in the slow-axis direction and a near-Gaussian distribution in the fast-axis direction (spot size of 1.5 × 6.6 mm). During all trials, the fast axis of the beam spot was set parallel to the traverse direction and the beam focal plane was positioned at the top surface of the test plates. Titanium powder was injected directly into the molten pool using an off-axis powder injection system. Argon was used as a shielding gas and was injected through a cylindrical nozzle (18 mm in diameter) at a flow rate of 8 l/min (coaxially with the powder stream). More details on the used laser surface alloying system are available elsewhere [22]. The laser alloying conditions used were selected based on the previous work [16,21] and are listed in Table 2. The alloying process was conducted for two extreme heat input (HI) levels (defined by the ratio of the laser power and the traverse speed) and traverse speeds, namely for the HI of 600 J/mm at a traverse speed of 3.33 mm/s (processing conditions no. 1, Table 2) and the HI of 1200 J/mm at a traverse speed of 1.25 mm/s (processing conditions no. 2, Table 2). As reported in the earlier work [16,21], the maximum amount of Ti which can be introduced into a molten pool, simultaneously ensuring compositional homogeneity of the alloyed bead, is determined by the HI level and laser power density. On the basis of these data, the Ti powder feed rates for a given HI level were selected to produce homogeneous alloyed beads with similar Ti contents. This made it possible to assess the effect of the cooling rate on the growth morphology of the TiC precipitates at a constant Ti concentration in the melt.

### 2.3. Examination of Thermal Conditions in the Molten Pool

The temperature profiles on the surface of the molten pool were recorded using an FLIR A 600-Series infrared camera (FLIR System, Inc. Wilsonville, OR, USA) with a measurement temperature range of 600 to 2000 °C and a maximum frame rate of 200 Hz. The infrared camera (IR) camara used had a spectral range of 7.5–14.0 µm. To estimate the emissivity of the melt in the molten pool, a two-step calibration procedure providing a direct correlation between the data of the IR and thermocouple measurements was used. During the calibration procedure, specially prepared test samples with titanium alloyed beads were heated by a laser beam under an argon atmosphere. The first step included IR measurements of temperature fields during the heating and melting of the test samples. In the second step, using the same heating conditions and test sample dimensions, the thermocouple measurements provided a pointwise check of the temperature fields previously recorded by the IR camera. An Agilent 34970A data logger (Agilent Technologies, Inc., Santa Clara, CA, USA) was used for the acquisition and recording of the thermocouple data. Additional calibration procedure details have been previously given [23].

### 2.4. Metallographic Examination

The microstructure of the alloyed beads was analysed by SEM and TEM. The SEM examination was performed using a ZEISS SUPRA 35 microscope (Carl Zeiss Microscopy GmbH, Jena, Germany) equipped with an energy-dispersive spectrometer (EDS, EDAX Inc., Mahwah, NJ, USA). For the SEM examination, the samples were polished through 0.04 μm colloidal silica and etched in 12% Nital. Area (volume) fractions of TiC and eutectic cementite phases were measured using the Nikon NIS-Elements Basic Research image processing software (ver. 3.13, Nikon Corporation, Tokyo, Japan). TiC phase measurements were conducted on backscattered electron (BSE) SEM images taken from nonetched samples. The mean free path (MFP) between the TiC precipitates was measured using the following equation:MFP = (1 − P_p_)/N_L_(1)
where P_p_ is a TiC fraction and N_L_ is the number of TiC precipitates intercepted per unit length of the test line [24].

Thin foils for TEM examination were prepared by a FIB micro-sampling technique using an FEI Helios NanoLab 600i SEM/FIB DualBeam microscope. The foils were examined under an FEI Titan 80/300 microscope (Scientific and Technical Instruments, Hillsboro, OR, USA). using conventional TEM as well as scanning TEM (STEM) imaging modes. The STEM observations were conducted with high-angle annular dark-field (HAADF) imaging. The phase composition of the alloyed beads and unit cell parameters of the TiC phase were determined using XRD. XRD data were acquired using a PANalytical X’Pert PRO MPD X-ray diffractometer (Malvern PANalytical, Malvern, UK) and a CoKα radiation (λ = 0.179 nm) source. The Rietveld method was used to quantify the austenite and ferrite (martensite) phases. In addition, phase and orientation maps were recorded by electron back-scatter diffraction (EBSD) techniques. The maps were created in the plane parallel to the alloyed bead surface (after grinding to a depth of 100 µm). EBSD analysis was conducted on a ZEISS SUPRA 35 SEM equipped with an OXFORD electron backscatter diffraction system.

### 2.5. Hardness and Nanoindentation Testing

Hardness profiles of the alloyed beads were determined using a Vickers Wilson Wolpert 401 MVD microhardness machine (Wilson Wolpert Instruments, Aachen, Germany). Hardness tests were performed on polished cross-sections of the bead using a 200 g load and a dwell time of 5 s. The average hardness of the matrix material in the alloyed bead was measured using a 25 g load applied over 10 s.

Nanoindentation testing was performed using a CMS Micro-Combi Tester (CSM Instruments, Needham, MA, USA) with a standard Bercovich probe. All tests were performed with the maximum load of 10 mN at a loading rate of 0.33 mN/s and a hold period of 10 s at maximum load. A total of 10 indents were averaged to determine the mean hardness (HIT) and elastic modulus (EIT) values for synthesised TiC precipitates.

## 3. Results

### 3.1. Examination of Thermal Conditions in the Molten Pool

The surface temperature profiles along the centre line of the molten pool recorded by IR thermography under processing conditions no. C1 and C2 (Table 2) are presented in Figure 1. The corresponding temperature profiles taken perpendicularly to the scanning direction at the laser beam spot location (approximately at the centre of the beam) are depicted in Figure 2. The maximum temperature of the molten pool was quite similar for both processing conditions and reached about 1750 °C. However, when comparing the temperature profiles along the molten pool (Figure 1), the temperature of the rear part of the molten pool was higher under processing conditions no. C1. This, in turn, led to approx. 2 times higher thermal gradients at the solidification front under processing condition no. C1 than under C2. The cooling rates at the trailing age of the molten pool were estimated to be approx. 700 ± 66 and 280 ± 41 °C/s under processing conditions nos. C1 and C2, respectively (in the temperature range of 1700–1300 °C, that is, before solidification of the matrix alloy occurs for the Fe-C-Si-Ti alloy system containing approximately 8.0 wt% of Ti [16]).

### 3.2. Macro- and Microstructure Analysis

The cross-sectional macrographs of alloyed beads produced under the investigated conditions are shown in Figure 3. The cross-sectional areas of the alloyed zone were measured to be 4.96 ± 0.43 and 7.24 ± 0.54 mm^2^ for processing conditions nos. C1 and C2, respectively. The corresponding depths of the alloyed zone were 1.23 ± 0.09 and 1.68 ± 0.10 mm, respectively. The above differences in the geometry of the alloyed zone are mainly attributed to the difference in heat input values for the investigated processing conditions. The typical microstructures that developed in the under-surface region of alloyed beads (~100 μm beneath the surface) are presented in Figure 4 and Figure 5. The morphologies of TiC precipitates are shown in Figure 6. The XRD patterns of both alloyed beads are shown in Figure 7. The chemical compositions and microstructural parameters of the alloyed beads are listed in Table 3 and Table 4, respectively.

Both processing conditions provided compositional homogeneity and a Ti content of approximately 8.0 wt% (Table 3). Microstructural characterisation indicated that the selected solidification conditions led to the formation of microstructures containing TiC precipitates, primary austenite grains partially transformed to martensite, and a ledeburite eutectic (Figure 5). Most of the TiC precipitates exhibited a dendritic growth morphology (at different development states). The lamellar morphology of the TiC phase, typical of the eutectic reaction, was not observed (Figure 6). Both beads exhibited a uniform distribution of TiC precipitates throughout the alloyed zone (Figure 4) and essentially the same fraction of the TiC phase (about 13 vol %; Table 4). However, the thermal conditions affected the size of the dendritic precipitates of the TiC phase. Under condition no. C1, most of the TiC precipitates showed early stages of dendritic growth morphology with sizes up to 10 µm (Figure 6a,b), whereas condition no. C2, associated with lower solidification and cooling rates, led to the formation of TiC dendritic precipitates with well-developed secondary arms and sizes up to 30 µm (Figure 6c,d). Moreover, quantitative metallography revealed that, despite the same fractions of TiC precipitates, the microstructure developed under processing conditions no. C1 exhibited a lower MFP than that formed under conditions no. C2. In the case of the processing conditions nos. C1 and C2, the MFPs were estimated to be 16 ± 4 and 24 ± 7 µm, respectively.

The microstructural data (Table 4) indicated that the change in the processing conditions did not affect the eutectic cementite fraction. In both cases, the volume fraction of eutectic cementite was approximately 17 vol %. However, it should be noted that the resulting change in thermal conditions has a pronounced influence on the phase composition of the matrix material. The increase in the cooling rate elevated the fraction of retained austenite. Based on a quantitative analysis of XRD patterns, the retained austenite fractions were estimated to be 32.9 ± 1.2 and 13.6 ± 1.0 wt% under processing conditions nos. C1 and C2, respectively. A large difference in the retained austenite fractions between the two alloyed beads was confirmed by the EBSD phase mapping presented in Figure 8. Note that Figure 8 includes EBSD orientation maps for the FCC phase to distinguish between the TiC and austenite phase. It is interesting to note that the EBSD phase mapping overestimated the cementite fraction in the microstructure developed under processing conditions no. 2, suggesting the presence of submicroscopic cementite precipitates in the austenitic/martensitic matrix (Figure 8b). TEM investigations confirmed the presence of nanometric precipitates of cementite in the matrix of the bead produced under processing conditions no. 2 (Figure 9b). The nanometric cementite precipitates had a predominantly spherical morphology and an average size of 100 nm (Figure 9b and Figure 10). The austenitic/martensitic matrix of the alloyed bead produced under processing conditions no. 1 did not contain submicroscopic cementite precipitates (Figure 9a).

### 3.3. Hardness Analysis

Figure 11 compares the hardness profiles of the alloyed beads. Note that a uniform hardness distribution confirms the uniform titanium concentration in the molten pool during processing. The average hardness values for alloyed beads produced under processing conditions nos. C1 and C2 were 663 ± 9 and 712 ± 10 HV0.2, respectively. The average hardness of the matrix material formed under these processing conditions was estimated to be 545 ± 41 and 606 ± 38 HV0.025, respectively.

Nanoindentation testing of the TiC precipitates indicated that the investigated thermal conditions in the molten pool had no effect on the mechanical properties of the synthesised TiC phase. Regardless of the thermal conditions in the molten pool, the average values of the hardness and elastic modulus were approx. 460 ± 22 and 45 ± 0.9 GPa, respectively. These values confirm the synthesis of the stoichiometric TiC phase under both investigated processing conditions [25].

## 4. Discussion

### 4.1. Examination of Thermal Conditions in the Molten Pool

Similar values of the maximum temperature in the molten pool under both processing conditions, despite significant differences in the HI level (600 and 1200 J/mm for processing conditions nos. C1 and C2, respectively), result from the unique combination of processing parameters. The selected processing conditions differ in both the traverse speed and the laser power. In turn, the laser power level for a given beam spot size, determines the laser power density. The traverse speed and laser power density are considered the most important variables that influence the temperature fields in the molten pool [26,27]. Generally, with increasing traverse speed, at given laser power density, the energy delivered to the processed surface becomes smaller because of the decrease in the laser beam interaction time. On the other hand, at low traverse speeds such as in the case of processing conditions no. C1 (1.25 mm/s), the maximum molten pool temperature is limited by the conduction heat loss. Significant impact on the maximum temperature on the molten pool formed under the processing parameters no. C1, that is under a notably higher traverse speed (3.33 mm/s) had a higher laser power density. Many works reported a direct influence of the laser power density on the peak temperature and the volume of the molten pool [28,29,30,31].

The difference in temperature profiles along the molten pool, affecting the temperature gradients at the trailing edge of the pool (Figure 1), is believed to be a combined effect of the higher traverse speed and the characteristics of fluid flow motion in the molten pool. Heat and mass transport in the molten pool during the laser surface alloying process is governed by the convective flow (Marangoni effect), which in turn is controlled by the temperature coefficient of the surface tension [32]. The cross-sectional micrographs in Figure 3 suggest that the surface tension temperature coefficient was positive, that is, the surface tension increases with increasing temperature. The positive value of the surface tension coefficient leads to inward surface fluid flow in the molten pool, which produces a downward flow at the centre of the molten pool carrying the hottest material to the bottom of the molten pool. This, in turn, induces outward flow at the bottom of the pool, providing heat transfer to the peripheral region of the pool, especially to the rear part of the molten pool (Figure 12) [33]. Furthermore, the above characteristic of Marangoni flow is supported by a high sulphur content in the melt (>0.100 wt%; Table 3) [34,35]. When comparing temperature profiles taken perpendicularly to scanning direction (Figure 2), larger thermal gradients existed on the pool surface under processing conditions no. C1. This suggests a higher intensity of fluid flow, which, in turn, elevated the melt temperature in the rear part of the molten pool. For this reason, the processing parameter no. C1, despite the notably higher traverse speed, led to higher thermal gradients at the trailing edge of the molten pool than conditions no. C2. As a consequence, the cooling rate (in the temperature range of 1700–1300 °C, that is, before the solidification of the matrix alloy) that existed at the trailing edge of the molten pool formed under processing conditions no. C1 was 2.5 times higher than that under processing conditions no. C2.

The high intensity of the fluid flow in the molten pool formed under processing parameters no. C1 is also responsible for the homogeneous distribution of TiC precipitates throughout the alloyed zone at a relatively high Ti concentration in the melt [16,21]. It should be noted that, despite the significant difference in the HI level, the Ti content was the same in both alloyed beads. This clearly shows a direct impact of the laser power density on the mass transport in the molten pool and the distribution of reinforcing TiC particles. Similar observations were reported by AlMangour et al. [36]. They processed TiC/stainless steel 316L nanocomposites by selective laser melting and reported that laser power density played an important role in intensifying the fluid flow in the molten pool and, consequently, in TiC distribution uniformity.

### 4.2. Microstructure and Hardness Analysis

Generally, the solidification reactions in the investigated multicomponent alloy are similar to those expected in the ternary Fe-Ti-C alloy system [21]. Figure 13 shows the Fe-rich corner of the Fe-Ti-C ternary phase diagram, with the marked point C representing the composition of the fabricated alloyed beads. For both alloyed beads, the first phase to precipitate in the melt is TiC (the primary TiC precipitates). Next, the solidification path intersects the monovariant eutectic line, separating the TiC and γ-Fe (austenite) phases, and follows the eutectic line resulting in simultaneous precipitation of the TiC and γ-Fe from the liquid by a monovariant eutectic reaction. Solidification terminates with the isothermal ternary eutectic reaction (e.g., L → TiC + γ-Fe + cementite).

Taking into account the data obtained from the examination of thermal conditions in the molten pool, it can be concluded that the cooling rates up to approximately 700 °C/s allow for a complete reaction between carbon and the entire amount of titanium introduced into the molten pool (at the optimal titanium concentration for the given HI level [16,26]). This finding is supported by the same TiC fraction in both alloyed beads and the fact that the TEM/EDS analysis did not detect the titanium presence in the matrix material of alloyed beads produced under processing parameters no. C1. Furthermore, the thermal data indicate that, in the case of the investigated Fe-based alloy melt containing approximately 8.0 wt% Ti, the cooling rates of about 700 and 280 °C/s (in the temperature range of 1700–1300 °C, that is, before the formation of the primary austenite phase—the matrix material) allow an equiaxed dendritic growth of TiC precipitations with developed secondary dendrite arms and sizes up to 10 and 30 µm, respectively (Figure 6). The increase in the size of the TiC precipitates, at the same TiC fraction in the beads, indicates a lower number of TiC precipitates for a given alloyed area. This was confirmed by the EBSD orientation maps presented in Figure 8c,d. This suggests that the change in the thermal conditions in the molten pool affected the nucleation process of the TiC phase. The dissolution of graphite nodules is believed to produce a large number of preferred sites for the nucleation of TiC precipitates (Figure 14). As a result, the Marangoni convection in the molten pool is responsible for the defragmentation of the graphite nodule and the distribution of the resulting nucleation sites throughout the melt. For this reason, a refinement of TiC precipitates in the microstructure developed under processing conditions no. C1 results from a combined effect of high solidification and cooling rates and also the intensive fluid flow in the molten pool.

The above-mentioned difference in size of TiC precipitates is directly responsible for the different values of the MFP. It has been reported that the MFP in homogenous composite structures, with the same fractions of reinforcing particles, increases with an increase in the size of the reinforcing particles [38].

The same lattice parameter for the TiC precipitates synthesised under both processing conditions (~0.433 nm—calculated from the XRD patterns) implies that the investigated thermal conditions in the molten pool did not affect the composition of the primary TiC phase. The value of the TiC lattice parameters indicates on the synthesizes of the stoichiometric TiC phase [39]. This is in good agreement with preciously mentioned results of nanoindentation testing.

A significant difference in the fraction of retained austenite in the synthesised composite structures (approx. 33 and 14 wt% for processing conditions nos. C1 and C2, respectively, Table 4) and the formation of submicroscopic cementite precipitates (under processing conditions no. C2) indicate that the cooling rate has a pronounced influence on the phase composition of the matrix material and consequently its mechanical properties. In general, the decrease in the cooling rate led to both a decrease in the fraction of retained austenite and the formation of nanometric cementite precipitates. For this reason, lower cooling rates provided a higher microhardness of the matrix material and, in the case of investigated Fe-Ti-C alloy, a higher total hardness of the produced TiC-reinforced composite structures.

An understanding of the formation mechanism of the observed nanometric cementite precipitates requires additional experiments, and will be presented in a separate paper.

## 5. Conclusions

The effect of the cooling rate in the molten pool on the microstructure of the in situ TiC-reinforced composite surface layer produced on ductile cast iron via laser surface alloying with titanium has been studied. Values of the cooling rate under two extreme heat input levels were estimated using infrared thermography. The results showed that the cooling rates from 280 to 700 °C/s did not affect the fraction of TiC reinforcing particles and provided a complete reaction between carbon and the entire amount of titanium introduced into the molten pool. However, the increase in the cooling rate led to a refinement of TiC reinforcing particles. For the melt containing 8.0 wt% of Ti, the cooling rate of 280 °C/s allowed the TiC particles (primary TiC phase) to grow in the dendritic form with well-developed secondary arms and a total size of up to 30 µm. At the higher cooling rate (700 °C/s), the size of the dendritic TiC particles was limited to 10 µm. The investigated conditions in the molten pool led to the synthesis of the stoichiometric primary TiC phase. Furthermore, it was found that the cooling rate influenced the phase composition of the matrix material. The increase in the cooling rate elevated the retained austenite fraction in the matrix material, lowering its hardness and consequently the overall hardness of the composite layers.

## Figures and Tables

**Figure 1 materials-17-00932-f001:**
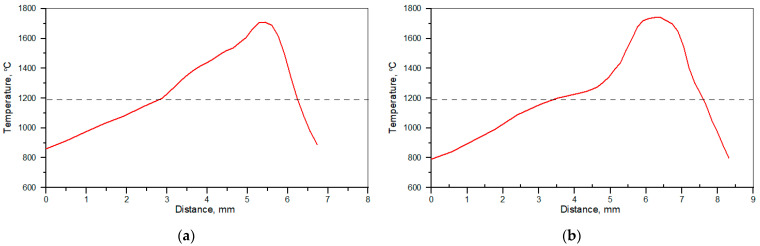
Temperature profiles at the molten pool surface along the centreline of the alloyed bead recorded under processing conditions nos. (**a**) C1 and (**b**) C2 (Table 2). The dashed lines indicate an approximate solidus temperature for the alloyed bead containing 8 wt% of Ti (determined using the thermodynamic calculations [16]).

**Figure 2 materials-17-00932-f002:**
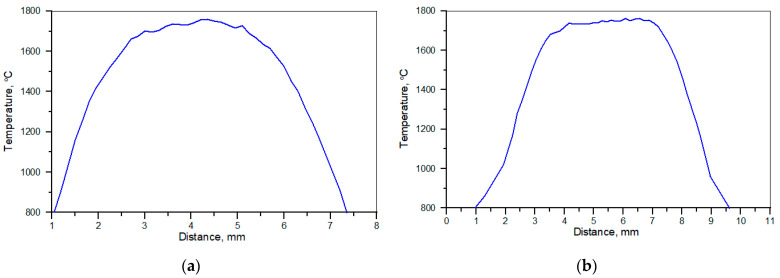
Temperature profiles at the molten pool surface taken perpendicularly to the scanning direction in the laser beam location. Processing conditions nos. (**a**) C1 and (**b**) C2 (Table 2).

**Figure 3 materials-17-00932-f003:**
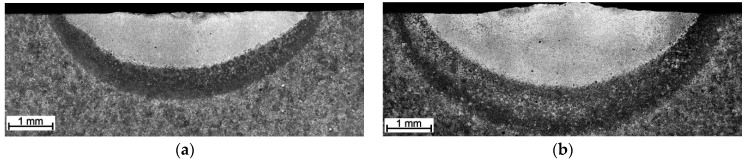
Optical macrographs of the traverse cross-section of the alloyed beads produced under processing conditions nos. (**a**) C1 and (**b**) C2 (Table 2).

**Figure 4 materials-17-00932-f004:**
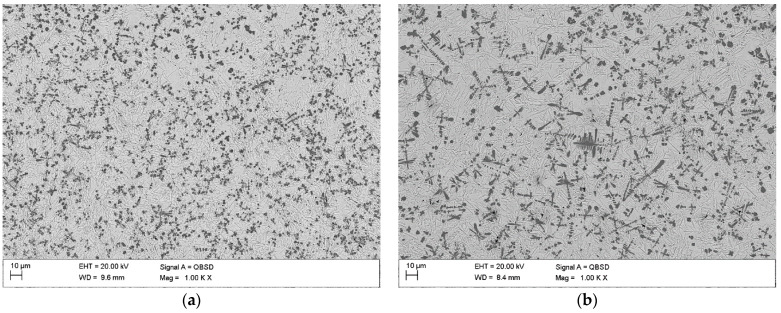
Low-magnification BSE SEM micrographs of the alloyed beads produced under processing conditions nos. (Table 2) (**a**) C1 and (**b**) C2 (images taken on the transverse cross-section of the bead approx. 100 μm beneath the bead’s surface).

**Figure 5 materials-17-00932-f005:**
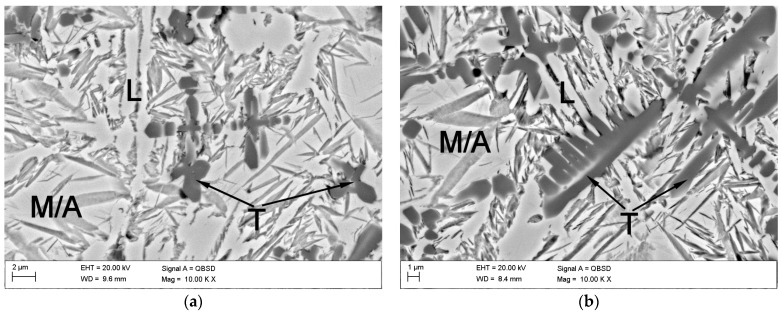
BSE SEM micrographs of the alloyed beads produced under processing conditions nos. (Table 2) (**a**) C1 and (**b**) C2 (images taken on the transverse cross-section of the bead approx. 100 μm beneath the bead’s surface; M/A—primary austenite grains partially transformed to martensite; L—ledeburite eutectic; T—primary TiC phase).

**Figure 6 materials-17-00932-f006:**
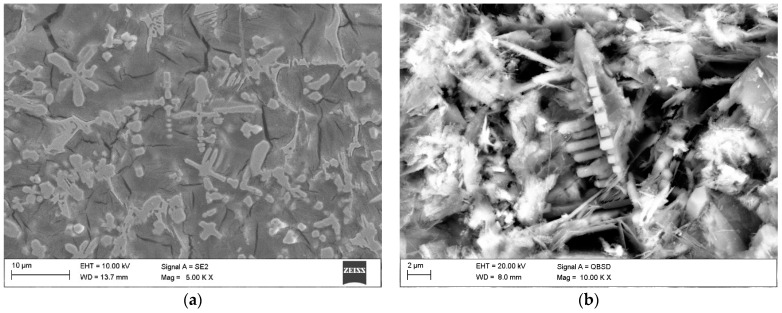
SEM deep-etched microstructures developed under processing conditions nos. (**a,b**) C1 and (**c,d**) C2; Table 2.

**Figure 7 materials-17-00932-f007:**
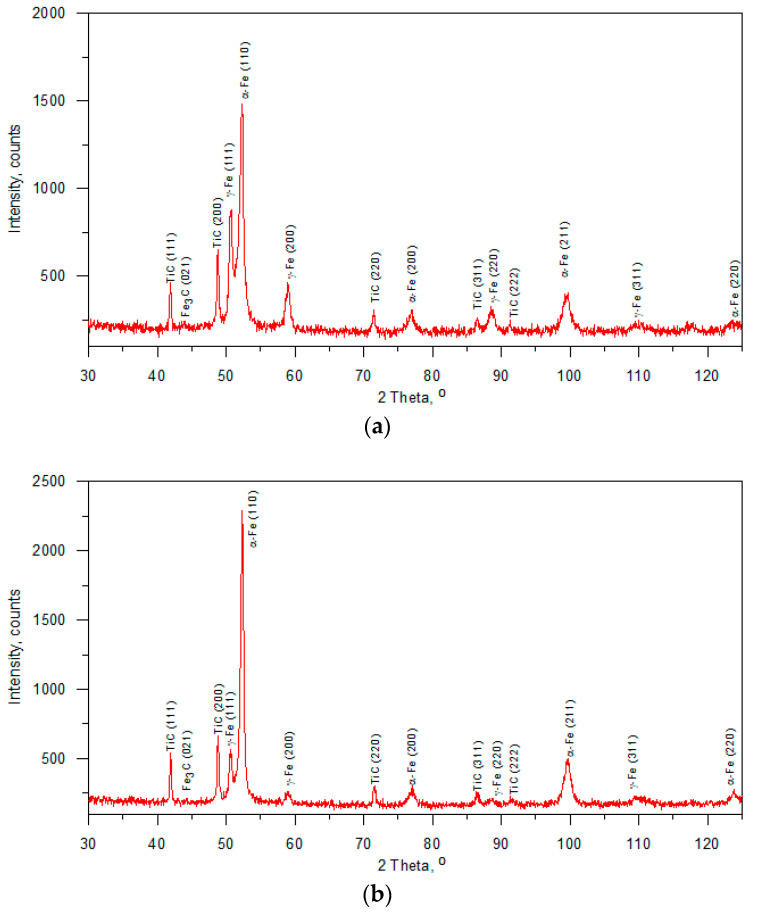
XRD patterns of microstructures developed under processing conditions nos. (**a**) C1 and (**b**) C2; Table 2.

**Figure 8 materials-17-00932-f008:**
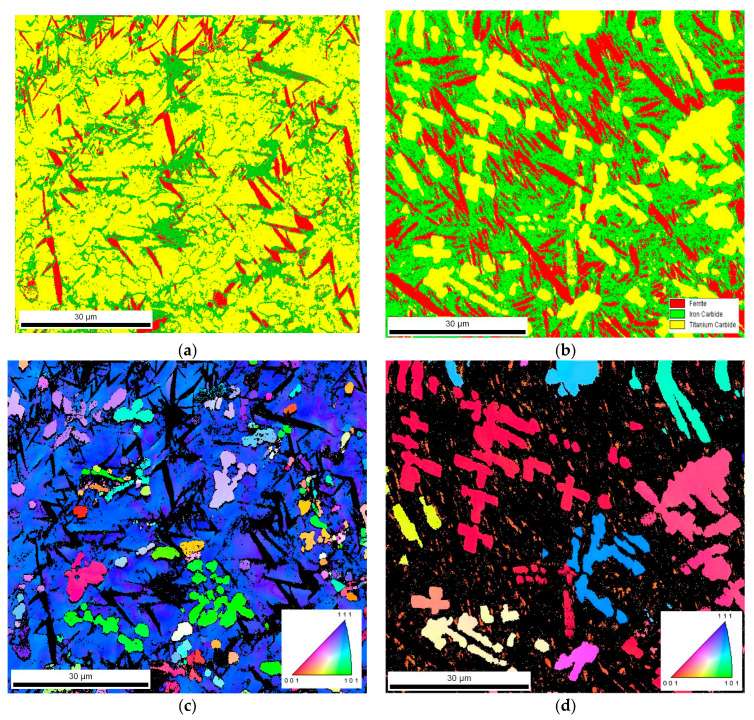
EBSD phase maps recorded on the surface of the alloyed bead produced under processing conditions nos. (**a**) C1 and (**b**) C2; (**c**,**d**) the corresponding EBSD orientation maps for TiC and austenite phase; Table 2.

**Figure 9 materials-17-00932-f009:**
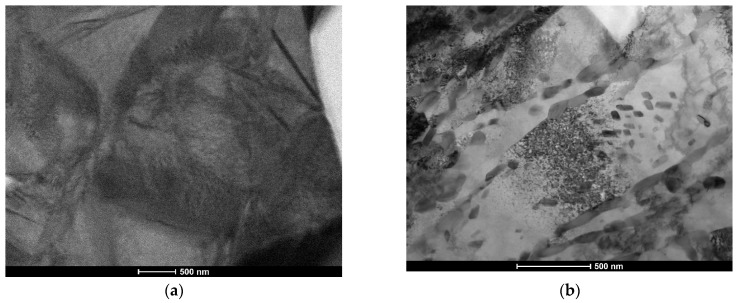
STEM HAADF images of the matrix material in alloyed beads produced under processing conditions nos. (**a**) C1 and (**b**) C2; Table 2.

**Figure 10 materials-17-00932-f010:**
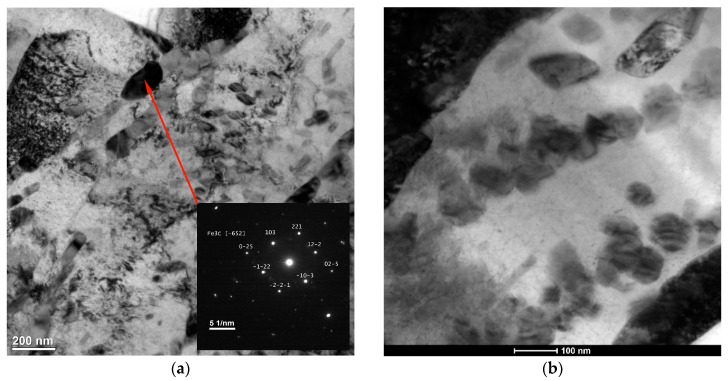
(**a**) Bright−field TEM and (**b**) STEM HAADF images showing the morphologies of nanometric cementite precipitates in the alloyed bead produced under processing conditions no. C2; Table 2. The insert in (**a**) is the SAED pattern obtained from the marked spherical precipitate.

**Figure 11 materials-17-00932-f011:**
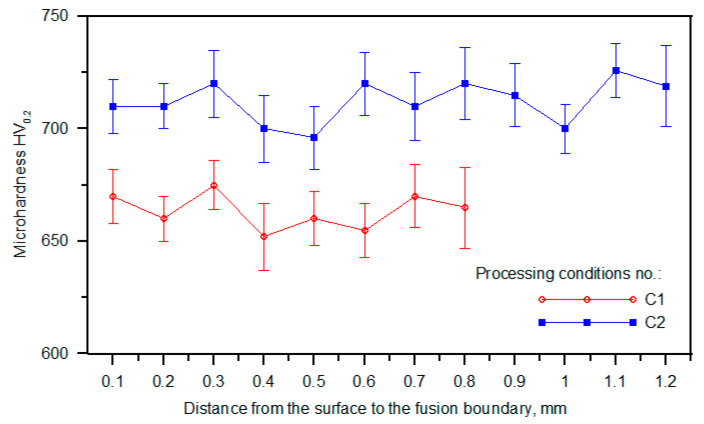
Hardness profiles of alloyed beads (Table 2).

**Figure 12 materials-17-00932-f012:**
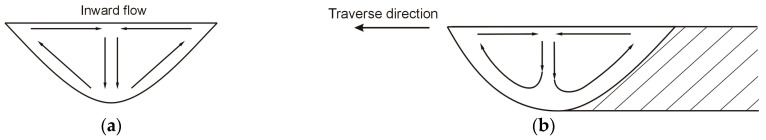
Schematic diagram of the flow in the molten pool caused by the positive temperature coefficient of surface tension: (**a**) cross-section; (**b**) longitudinal section [33].

**Figure 13 materials-17-00932-f013:**
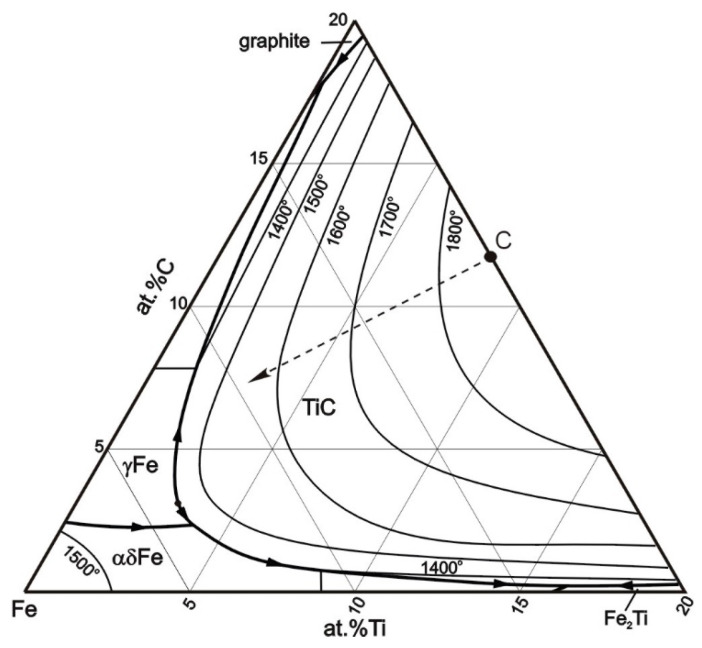
The liquidus projection of the Fe-Ti-C phase diagram in the Fe-rich corner. Point C indicates the chemical composition of the fabricated alloyed beads. Reproduced from [37] with the permission of publisher.

**Figure 14 materials-17-00932-f014:**
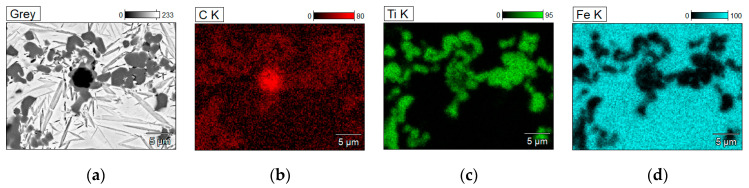
(**a**) SEM image showing the formation of TiC precipitates in the direct vicinity of a partially dissolved graphite nodule; (**b**–**d**) corresponding maps of C, Ti, and Fe distribution, respectively.

**Table 1 materials-17-00932-t001:** Chemical composition of the DCI test plates (wt%).

C	Si	Cu	Mn	Cr	Ni	Ti	S	P	Fe
3.61	2.50	0.81	0.24	0.03	0.03	0.02	0.008	0.021	balance

**Table 2 materials-17-00932-t002:** Selected processing parameters of the LSA process.

Processing Conditions No.	Laser Power (W)	Traverse Speed (mm/s)	Heat Input ^1^ (J/mm)	Powder Feed Rate (g/min)
C1	2000	3.33	600	1.2
C2	1500	1.25	1200	0.7

^1^ Defined by the ratio of the laser power and the traverse speed.

**Table 3 materials-17-00932-t003:** Chemical composition of the alloyed beads (wt%).

Processing Conditions No.	C	Ti	Si	Mn	Cu	Ni	S	P	Fe
C1	3.11	7.9	2.04	0.160	0.64	0.03	0.125	0.020	balance
C2	3.16	8.2	2.03	0.166	0.79	0.03	0.131	0.020	balance

**Table 4 materials-17-00932-t004:** Microstructural parameters of the alloyed beads.

Processing Conditions No.	α-Fe (Martensite) Fraction (wt%)	Retained Austenite Fraction (wt%)	Eutectic Cementite Fraction (vol %)	TiC Fraction (vol %)
C1	37.6 ± 1.2	32.9 ± 1.2	17.2 ± 2.2	12.9 ± 1.0
C2	67.8 ± 1.5	13.6 ± 1.0	16.4 ± 2.1	13.2 ± 1.1

## Data Availability

Data are contained within the article.

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
