# Peer review of "Effect of Cooling Rate on Microstructure of In Situ TiC-Reinforced Composite Surface Layers Synthesized on Ductile Cast Iron by Laser Alloying"

_materials, 2024, doi:10.3390/ma17040932_

Round 1

Reviewer 1 Report

Comments and Suggestions for Authors

The authors investigate the effect of cooling rate on microstructure of in-situ TiC-reinforced composite surface layers synthesized on ductile cast  iron by laser alloying. However, we cannot see the in-situ observation in the study. Thus, it is not necessary to use the specific term of in-situ. 

In this work they use an infrared camera to measure and record the surface temperature of the molten pool, but they provide a raw data of the temperature distribution map on the molten pool surface.

In microstructural characterization they show martensite (partially or fully) and a ledeburite structure in the interdendritic regions in Figure 5. It will be better to indicate the corresponding positions in Figure 5. The same case is found in the presence of submicroscopic cementite precipitates in Figure 8b.

The Figure 12 is cited from [33] but it is not cited in Figure 12.

They don't make a conclusion at the end to summarize some key points of the results in this work.

Author Response

RESPONSE TO REVIEWER 1

Thank you for your thorough review of our manuscript. We have corrected the manuscript considering the comments, and We provided the required explanations. 

Responses to your comments and suggestions:

Comment 1:  “The authors investigate the effect of cooling rate on microstructure of in-situ TiC-reinforced composite surface layers synthesized on ductile cast  iron by laser alloying. However, we cannot see the in-situ observation in the study. Thus, it is not necessary to use the specific term of in-situ.

Response: The term “in situ” refers directly to the method of composite structure formation. During the laser alloying process of the cast irons, the composite surface layers can be achieved using both the ex-situ and in-situ methods. The ex situ method involves the introduction of the reinforcing particles, synthesised in the separated process, into the molten pool. In the case of the in situ method, the reinforcing particles (carbide precipitates) are formed within the molten pool by a reaction between the carbide and elements of a strong carbide-forming tendency introduced during the process.

Comment 2:In this work they use an infrared camera to measure and record the surface temperature of the molten pool, but they provide a raw data of the temperature distribution map on the molten pool surface.”

Response: The IR measurements were focused directly to achieve the temperature profiles in the molten pool and to estimate the values of the cooling rate. Please note that to achieve the presented temperature profiles, we conducted the two-step calibration procedure (we have to estimate the emissivity values). The elaborated temperature profiles allowed us to show the main differences in the thermal conditions in the molten pool between the two analysed processing conditions.

Comment 3:In microstructural characterization they show martensite (partially or fully) and a ledeburite structure in the interdendritic regions in Figure 5. It will be better to indicate the corresponding positions in Figure 5. The same case is found in the presence of submicroscopic cementite precipitates in Figure 8b.”

Response: The locations of the analysed phases have been marked in the SEM images (Figure 5). In the case of Figure 8b, the potential locations of submicroscopic cementite precipitates are indicated by the iron carbide region (green area-map legend).

Additionally, we slightly modified the microstructure description in the text as follows:

Microstructural characterization indicated that the selected solidification conditions led to the formation of microstructures containing TiC precipitates, primary austenite grains partially transformed to martensite and a ledeburite eutectic (Figure 5).

Comment 4:The Figure 12 is cited from [33] but it is not cited in Figure 12.”

Response: The reference [33] has been added to the Figure 12.   

Comment 5:They don't make a conclusion at the end to summarize some key points of the results in this work.”

Response: We inserted the following Conclusions Section:

The effect of cooling rate in the molten pool on the microstructure of the in situ TiC-reinforced composite surface layer produced on the ductile cast iron via a laser surface alloying with titanium has been studied. Values of the cooling rate under two extreme heat input levels were estimated using infrared thermography. The results showed that the cooling rates from 280 to 700 °C/s did not affect the fraction of TiC reinforcing particles and provided a complete reaction between carbon and the entire amount of titanium introduced into the molten pool. However, the increase in the cooling rate led to a refinement of TiC reinforcing particles. For the melt containing 8.0 wt% of Ti, the cooling rate of 280 °C/s allowed the TiC particles (primary TiC phase) to grow in the dendritic form with well-developed secondary arms and a total size of up to 30 µm. At the higher cooling rate (700 °C/s), the size of the dendritic TiC particles was limited to 10 µm. The investigated conditions in the molten pool led to the synthesis of the stoichiometric primary TiC phase. Furthermore, it was found that the cooling rate influenced the phase composition of the matrix material. The increase in the cooling rate elevated the retained austenite fraction in the matrix material, lowering its hardness, and consequently the overall hardness of the composite layers.

Reviewer 2 Report

Comments and Suggestions for Authors

- A good and detailed analysis of microstructure, phases and solidifications process is done for the produced beads. For a surface coating many beads must be aplied side by side . What is the spected microstructure and phases alterations for many beads? The microstructure and properties can be considerably changed?

- Some discussion about spected wear resistance of both evaluated process conditions must be added; 

- The conclusions chapter should be added

Comments on the Quality of English Language

No comments.

Author Response

RESPONSE TO REVIEWER 2

Thank you for your thorough review of our manuscript. We have corrected the manuscript considering the comments, and We provided the required explanations. 

Responses to your comments and suggestions:

Comment 1:  “A good and detailed analysis of microstructure, phases and solidifications process is done for the produced beads. For a surface coating many beads must be aplied side by side.

and

What is the spected microstructure and phases alterations for many beads? The microstructure and properties can be considerably changed?”

Response: Because of the high thermal stability of TiC phase, the TiC precipitates are generally not affected by the thermal cycle of the consecutive bead in the overlap boundary between the beads during overlapping multi-bead alloying process. As a results, the TiC distribution is uniform throughout the surface layer. Furthermore, the morphology and size of TiC precipitates are stable and defined by the solidification conditions in the molten pool. In contrast the matrix material is prone to phase transformations in the overlap boundary between consecutive beads. Depending on the processing conditions, the cementite phase undergoes dissolution in the austenite phase, leading to the formation of the austenite zone (containing high C content) and even graphitisation in the heat affected zone between the consecutive beads. These phenomena directly influence mechanical properties and are very important from a practical point of view. They are reported in the earlier works:

Janicki, D.  Fabrication of TiC-Reinforced Surface Layers on Ductile Cast Iron Substrate by Laser Surface Alloying. Solid. State Phenom. 2020, 308, 76-99.

Janicki, D. Effect of Chromium and Molybdenum Addition on the Microstructure of In Situ TiC-Reinforced Composite Surface Layers Fabricated on Ductile Cast Iron by Laser Alloying. Materials 2020, 13, 5750.

Comment 2:  “Some discussion about spected wear resistance of both evaluated process conditions must be added;” 

Response:  We would like to present the wear behaviour of the investigated composite surface layers in the future manuscript after comprehensive wear testing.

Comment 3:  “The conclusions chapter should be added.”

Response: We inserted the following Conclusions Section:

The effect of cooling rate in the molten pool on the microstructure of the in situ TiC-reinforced composite surface layer produced on the ductile cast iron via a laser surface alloying with titanium has been studied. Values of the cooling rate under two extreme heat input levels were estimated using infrared thermography. The results showed that the cooling rates from 280 to 700 °C/s did not affect the fraction of TiC reinforcing particles and provided a complete reaction between carbon and the entire amount of titanium introduced into the molten pool. However, the increase in the cooling rate led to a refinement of TiC reinforcing particles. For the melt containing 8.0 wt% of Ti, the cooling rate of 280 °C/s allowed the TiC particles (primary TiC phase) to grow in the dendritic form with well-developed secondary arms and a total size of up to 30 µm. At the higher cooling rate (700 °C/s), the size of the dendritic TiC particles was limited to 10 µm. The investigated conditions in the molten pool led to the synthesis of the stoichiometric primary TiC phase. Furthermore, it was found that the cooling rate influenced the phase composition of the matrix material. The increase in the cooling rate elevated the retained austenite fraction in the matrix material, lowering its hardness, and consequently the overall hardness of the composite layers.

Reviewer 3 Report

Comments and Suggestions for Authors

In this article, several critical and minor issues require extensive revision from the authors. I cannot recommend publishing the current version of this article.

(1) Line 178, please add a detailed comparative analysis of the two pictures in Figure 3.

(2) Article missing title " 5. Conclusions ", please check and add.

(3) In Table 1, the lower frame line of the second row of the table should be 1.5 pounds, please check and modify.

(4) Lines 250-251, it should be described as " The average hardness of matrix material... ".

(5) Line 364, the expression is not clear. Please add metallographic data comparison here.

(6) In Figure 7, the picture is not clear.

(7) Line 112, add the basic steps for " a two-step calibration procedure ".

(8) Lines 162-163, in " the temperature of the rear part of the molten pool was higher under processing condition no.1 ", how much higher than " processing condition no.2 " ?

Author Response

RESPONSE TO REVIEWER 3

Thank you for your thorough review of our manuscript. We have corrected the manuscript considering the comments, and We provided the required explanations. 

Comment 1: “Line 178, please add a detailed comparative analysis of the two pictures in Figure 3”.

Response: The following sentences have been added:

Cross-sectional macrographs of alloyed beads produced under the investigated conditions are shown in Figure 3.  The cross-sectional areas of the alloyed zone were measured to be 4.96±0.43 and 7.24±0.54 mm2 for processing conditions no. C1 and C2, respectively. The corresponding depths of the alloyed zone were 1.23±0.09 and 1.66±10 mm, respectively. The above differences in the geometry of the alloyed zone are mainly attributed to the difference in heat input values for investigated processing conditions.

Comment 2: “Article missing title " 5. Conclusions ", please check and add.”

Response: We inserted the following Conclusions Section:

  1. Conclusions

The effect of cooling rate in the molten pool on the microstructure of the in situ TiC-reinforced composite surface layer produced on the ductile cast iron via a laser surface alloying with titanium has been studied. Values of the cooling rate under two extreme heat input levels were estimated using infrared thermography. The results showed that the cooling rates from 280 to 700 °C/s did not affect the fraction of TiC reinforcing particles and provided a complete reaction between carbon and the entire amount of titanium introduced into the molten pool. However, the increase in the cooling rate led to a refinement of TiC reinforcing particles. For the melt containing 8.0 wt% of Ti, the cooling rate of 280 °C/s allowed the TiC particles (primary TiC phase) to grow in the dendritic form with well-developed secondary arms and a total size of up to 30 µm. At the higher cooling rate (700 °C/s), the size of the dendritic TiC particles was limited to 10 µm. The investigated conditions in the molten pool led to the synthesis of the stoichiometric primary TiC phase. Furthermore, it was found that the cooling rate influenced the phase composition of the matrix material. The increase in the cooling rate elevated the retained austenite fraction in the matrix material, lowering its hardness, and consequently the overall hardness of the composite layers.

Comment 3: “In Table 1, the lower frame line of the second row of the table should be 1.5 pounds, please check and modify.”

Response: It has been modified.

 Comment 4: “Lines 250-251, it should be described as " The average hardness of matrix material... ”

Response: It has been corrected.

 Comment 5: “Line 364, the expression is not clear. Please add metallographic data comparison here.”

Response: We modified the sentence as follow:

A significant difference in the fraction of retained austenite in the synthesised composite structures (approx. 33 and 14 wt % for processing conditions no. C1 and C2, respectively, Table 4) and the formation of submicroscopic cementite precipitates (under processing conditions no. C2) indicate that the cooling rate has a pronounced influence on the phase composition of the matrix material and consequently its mechanical properties.

Comment 6: “In Figure 7, the picture is not clear.”

Response: The pictures of the XRD patterns have been enlarged and presented in higher quality.

Comment 7: “Line 112, add the basic steps for " a two-step calibration procedure ".

Response: The following description has been added:

To estimate the emissivity of the melt in the molten pool, a two-step calibration procedure providing a direct correlation between the data of the IR and thermocouple measurements was used. During the calibration procedure, specially prepared test samples with titanium alloyed beads were heated by laser beam under an argon atmosphere. The first step included IR measurements of temperature fields during the heating and melting of the test samples. In the second step, using the same heating conditions and test samples’ dimensions, the thermocouple measurements provided a pointwise check of the temperature fields previously recorded by the IR camera. An Agilent 34970A data logger (Agilent Technologies, Inc., Santa Clara, CA, USA) was used for the acquisition and recording of the thermocouple data. Additional calibration procedure details have been previously given [23].

Comment 8: “Lines 162-163, in " the temperature of the rear part of the molten pool was higher under processing condition no.1 ", how much higher than " processing condition no.2 " ?".

Response: To provide more detailed information on the temperature difference at the rear part of the molten pool, we modified the paragraph, as follows:

However, when comparing the temperature profiles along the molten pool (Figure 1), the temperature of the rear part of the molten pool was higher under processing condition no. C1. This, in turn, led to approx. 2-times higher thermal gradients at the solidification front under processing conditions no. C1 than that under C2.
